# Specific Six-Transmembrane Epithelial Antigen of the Prostate 1 Capture with Gellan Gum Microspheres: Design, Optimization and Integration

**DOI:** 10.3390/ijms24031949

**Published:** 2023-01-18

**Authors:** João Batista-Silva, Diana Gomes, Jorge Barroca-Ferreira, Eugénia Gallardo, Ângela Sousa, Luís A. Passarinha

**Affiliations:** 1CICS-UBI–Health Sciences Research Centre, University of Beira Interior, 6201-506 Covilhã, Portugal; 2Associate Laboratory i4HB–Institute for Health and Bioeconomy, NOVA School of Science and Technology, Universidade NOVA de Lisboa, 2829-516 Caparica, Portugal; 3UCIBIO–Applied Molecular Biosciences Unit, Department of Chemistry, NOVA School of Science and Technology, Universidade NOVA de Lisboa, 2829-516 Caparica, Portugal; 4Laboratório de Fármaco-Toxicologia–UBIMedical, University of Beira Interior, 6201-284 Covilhã, Portugal

**Keywords:** STEAP1, Gellan gum microspheres, batch method, capture, co-immunoprecipitation

## Abstract

This work demonstrates the potential of calcium- and nickel-crosslinked Gellan Gum (GG) microspheres to capture the Six-Transmembrane Epithelial Antigen of the Prostate 1 (STEAP1) directly from complex *Komagataella pastoris* mini-bioreactor lysates in a batch method. Calcium-crosslinked microspheres were applied in an ionic exchange strategy, by manipulation of pH and ionic strength, whereas nickel-crosslinked microspheres were applied in an affinity strategy, mirroring a standard immobilized metal affinity chromatography. Both formulations presented small diameters, with appreciable crosslinker content, but calcium-crosslinked microspheres were far smoother. The most promising results were obtained for the ionic strategy, wherein calcium-crosslinked GG microspheres were able to completely bind 0.1% (*v*/*v*) DM solubilized STEAP1 in lysate samples (~7 mg/mL). The target protein was eluted in a complexed state at pH 11 with 500 mM NaCl in 10 mM Tris buffer, in a single step with minimal losses. Coupling the batch clarified sample with a co-immunoprecipitation polishing step yields a sample of monomeric STEAP1 with a high degree of purity. For the first time, we demonstrate the potential of a gellan batch method to function as a clarification and primary capture method towards STEAP1, a membrane protein, simplifying and reducing the costs of standard purification workflows.

## 1. Introduction

Prostate cancer (PCa) is the second most frequently occurring cancer in males worldwide. According to Globocan, PCa is predicted to rise and become the most prevalent malignancy in males in 2040 with upwards of 2.4 million new cases per year [1]. Indeed, PCa is diagnosed through the levels of prostate-specific antigen (PSA). However, PSA often fails to distinguish between PCa and benign prostatic hyperplasia or prostatitis, giving rise to false positives [2,3]. Current treatment options, such as prostatectomy, androgen ablation, radiation and chemotherapy, seem to work for early stage PCa. Nevertheless, when PCa progresses to an androgen-independent metastatic phase or in the case of biochemical recurrence, these treatments become largely ineffective and low overall survivability is observed, making new approaches an imperative necessity [4,5].

The Six-Transmembrane Epithelial Antigen of the Prostate 1 (STEAP1), first discovered in 1999 [2], is a membrane protein overexpressed in PCa, while being mostly absent from other tissues or vital organs [2,6]. Due to the secondary structure of STEAP1 and localization at the cell surface in tight- and gap-junctions, it has been suggested to function as a transmembrane channel, transporting ions and small molecules, while also playing a role in cell adhesion and intercellular communication [5,6,7,8]. Further, when associated in heterotrimers with other STEAP family members, it seems to have metal reductase functions, being involved in the reduction and uptake of iron and copper [9,10,11]. Moreover, STEAP1 has been linked with oxidative stress responses and elevated levels of reactive oxygen species, which in turn, activate redox-sensitive and pro-invasive genes [12]. In addition, STEAP1 overexpression has been suggested to be a driving force for tumor initiation and progression [3]. Overall, STEAP1 seems to enhance tumor proliferation and aggressiveness, making it a potential PCa biomarker and therapeutic target. Indeed, STEAP1 has been appointed as a tumor-associated antigen that can function as a target for immunotherapy. In fact, [89Zr]Zr-DFO-MSTP2109A anti-STEAP1 antibody proved to be well tolerated and adequate for positron emission tomography bioimaging in PCa, being able to track changes in STEAP1 expression, and consequently, tracking the progression of PCa [13,14,15]. Further, the conjugation of anti-STEAP1 antibodies with monomethyl auristatin E, a potent antimitotic agent, has shown potential in reducing tumor volume and delaying castration-resistant PCa [16,17,18]. Another emerging strategy is the priming of cytotoxic T lymphocytes with STEAP1-derived epitopes for enhanced immune response. This approach has been associated with higher T cell infiltration in the tumor microenvironment [19], reduced metastases [20] and tumor inhibition [21]. Although promising, the development of immunotherapies is in its early days, and the efficacy is modest, being only a matter of time until the immunosuppressive nature of the tumor microenvironment rejects these therapeutics. It has also been shown that STEAP1 directly contributes to this immunosuppression [22]. Furthermore, although a STEAP1 structure has been deposited in the Protein Data Bank (PDB; https://www.rcsb.org/; accessed on 28 September 2022) with the accession code 6Y9B [10], it is incomplete, lacking both N- and C-termini. Indeed, several sites have been predicted in both these domains for post translational modifications (PTM) [23]. PTMs have also been proposed as one of the major differences between non-neoplastic PNT1A and neoplastic LNCaP cells, the latter being far more stable [23]. Thus, a complete crystallized STEAP1 structure is mandatory, both for in silico modeling, as well as to potentially increase the effectiveness of current immunotherapy approaches through better structural understanding. However, current purification approaches are scarce and seem to be mostly based on sequential chromatographic steps of immobilized metal affinity chromatography (IMAC) and size exclusion chromatography (SEC) [9,10]. This purification workflow has shown promise in the crystallization of other membrane proteins [24]. For instance, stable crystals were obtained from lysine-specific permease [25] and CdsD [26] extracts purified by IMAC and SEC in preliminary X-ray diffraction studies. Nonetheless, in the case of STEAP1, this approach appears to not be yielding enough protein concentration for crystallization studies, prompting the development of new isolation bioprocesses.

Gellan Gum (GG) is a natural linear anionic exopolysaccharide secreted by *Sphingomonas paucimobilis*, which consists of four repeating carbohydrates, including two β-D-glucoses, one α-L-rhamnose, and one β-D-glucuronic acid [27,28]. Due to its properties of biocompatibility, biodegradability, hydrophilicity, mucoadhesive features and good gelling capacity, GG has found remarkable success in the fields of food [29], tissue engineering [30], bioremediation [31], biosynthesis [32] and drug delivery [33]. Indeed, GG-based materials have been shown to promote strong adsorption of small drug molecules [34]. Recently, our research group demonstrated that GG microspheres can efficiently capture soluble catechol-O-methyltransferase (COMT) [35] and plasmid DNA [36]. Current microsphere-based methods being developed for protein capture are mostly based on magnetic microspheres, which quickly enhance the complexity and cost of bioprocesses [37,38,39,40]. Contrarily, GG microspheres are cost-effective and production methods are easier to scale up [41]. Further, and until now, protein capture mediated through microparticles seems to be solely restricted to soluble proteins. Certainly, the difficulties associated with membrane protein purification, such as loss of stability and natural conformation, act as a deterrent for new capture procedures [42]. In fact, out of 195,858 structures deposited in PDB, only 10,229 correspond to membrane proteins (accessed on 28 September 2022). Nonetheless, membrane proteins play a pivotal role in biological processes and emerging capture, and purification strategies should be explored towards structural determination.

Considering the necessity for novel bioprocesses and the efficacy previously demonstrated by GG microspheres in the capture of other biomolecules, the main purpose of the present work was to explore the potential of GG microspheres to capture recombinant human STEAP1, a highly relevant membrane protein, from *Komagataella pastoris* lysates, through a simple batch method. To achieve this, GG microspheres were reinforced with calcium and nickel ions, and two different approaches were studied. For calcium-crosslinked GG an ionic exchange strategy was conducted by the manipulation of pH and ionic strength. For nickel-crosslinked GG, an affinity approach was performed, mirroring IMAC.

## 2. Results

### 2.1. Characterization of Gellan Gum Microspheres

GG microspheres produced through W/O emulsion were crosslinked with calcium and nickel ions. Calcium was chosen as it is the most widely used crosslinker for GG microspheres by researchers, with proven efficacy in drug delivery [43], immobilization of cells and enzymes [32,44] and degradation of pollutants [31]. Nickel was selected since our research group had previously demonstrated this ion yielded the best capture and purification results for COMT, through a similar GG batch method [35]. Indeed, because nickel-crosslinked GG microspheres had already been described elsewhere, they were excluded from further characterization. Calcium-crosslinked GG microspheres were characterized in regard to size, morphology and elemental composition, through semi-optical microscopy, SEM, EDX and FTIR.

The mean diameter for calcium-crosslinked GG microspheres was attained through the average of six (n = 6) snapshots from semi-optical microscopy. The obtained mean diameter was of 330.37 ± 11.38 µm. Nickel-crosslinked GG microspheres had been previously described with a mean diameter of 239.06 ± 5.43 µm [35]. Representative snapshots can be seen in Figure 1.

Further, the morphology and geometry of calcium-crosslinked microspheres was assessed through SEM. The microspheres present a consistent and uniform structure with spherical shape. Nickel-, magnesium- and copper-crosslinked GG microspheres had also been previously reported with a spherical shape. However, the aforementioned microspheres presented clear rugosity, cavities and irregularities in their surface [35,36]. By comparison, calcium-crosslinked GG microspheres are far smoother, with no apparent pores, cavities or cracks (Figure 1D). Following SEM, microspheres were analyzed by EDX to unveil the main elemental composition. A summary of the results can be found in Table 1. As previously mentioned, GG is mainly comprised of carbohydrates, which validates that the major chemical elements in the microspheres are carbon and oxygen. Further, calcium was detected at an appreciable level in calcium-crosslinked GG microspheres, confirming appropriate crosslinking. When compared to nickel-crosslinked GG microspheres, the normalized ion percentage levels seem to differ nearly two-fold. Indeed, when comparing copper-, magnesium-, nickel- and calcium-crosslinked microspheres by crosslinker percentage, it seems that transition metals are incorporated in higher degrees than alkaline earth metals [35,36]. This stronger crosslinker concentration can induce the formation of a tighter mesh network, resulting in more compact microspheres, which can justify why nickel-crosslinked microspheres are smaller than calcium-crosslinked microspheres [45].

Then, FTIR analysis was performed to evaluate the chemical integrity of GG after microsphere assembly, as well as to detect the chemical interactions between GG and calcium. The recorded FTIR spectra for both GG powder and calcium-crosslinked GG microspheres can be seen in Figure 2. The spectrum of GG powder showed characteristic peaks at 3333 cm^−1^, due to the stretching of hydroxyl groups (-OH) from glucopyranose rings. The peak at 2912 cm^−1^ is assigned to -CH vibrations [46,47]. Further peaks at 1605 cm^−1^ and 1400 cm^−1^ correspond to the presence of carboxylate anions (COO^−^). The peak at 1026 cm^−1^ is linked to hydroxylic C-O stretching [46,47]. The spectrum for calcium-crosslinked GG microspheres displays similar peaks, although with slight variations in absorbance. Indeed, the rise of a peak at 1743 cm^−1^ and the disappearance of the peak at 1400 cm^−1^ suggests an interaction between the carboxyl groups from GG with calcium ions. Further, the quenching of the peaks at 3333 cm^−1^ and 1026 cm^−1^ might suggest that calcium could also interact with the glucopyranose rings of glucose and with the negatively charged components of glucuronic acid, respectively. It appears that all subunits of GG are involved in the coordination of calcium binding and this change in FTIR spectra corroborates EDX results, confirming calcium crosslinking.

### 2.2. Optimization of the Batch Method for the Capture of STEAP1

As previously mentioned, the batch method employed follows a simple sequence of binding, washing and elution steps. It was intended to take advantage of the high predicted STEAP1 isoelectric point of approximately 9.2 (Compute pI/Mw–Expasy; https://web.expasy.org/compute_pi/; accessed on 11 January 2022) to separate it from the remainder *K. pastoris* proteome with an average isoelectric point of 6.46 (Proteome-pI database; [48]). Although *K. pastoris* X33 Mut^+^ were used for STEAP1 production, instead of the listed *K. pastoris* strain GS115 in the Proteome-pI database, no significant changes were expected in isoelectric point since X33 is derived from GS115 [49]. So, the explored initial strategy was an ionic exchange, based solely on pH manipulation for both microspheres, where it was intended to bind STEAP1 to GG microspheres at pH 6.2 in 10 mM MES buffer, wash off most impurities at pH 8 in 10 mM Tris buffer and then start eluting STEAP1 at pH 9.2 or higher in 10 mM Tris buffer, either by charge neutralization or charge repulsion. The expected molecular weight of the recombinant STEAP1 produced by *K. pastoris* in mini-bioreactor cultures is ~35, or ~48 and ~63 kDa due to some aggregation events. However, as can be seen in Figure 3, protein samples recuperated from the initial batch with both microsphere crosslinkers presented an elevated molecular weight of >245 kDa. Indeed, it appears that STEAP1 tended to form complexes with GG microspheres, and in turn, present high molecular weight aggregates. It had been previously reported that sample boiling prior to Western blot for other transmembrane proteins resulted in similar large molecular weight complexes in immunoreactive assays [50]. To evaluate this condition, batch samples were left at room temperature, whereas their equivalent counterparts were boiled at 100 °C for 5 min prior to detection, however no change was detected for either condition (data not shown). Therefore, a series of optimizations were conducted to improve protein stability and solubility, namely, detergent solubilization, initial lysate concentration screening and microspheres volume ratios.

First, and since solubilization of membrane proteins is of the utmost importance for proper stabilization and conformation outside the natural lipidic environment [51,52], several mild nonionic detergents that our group had previously tested for STEAP1 at a 0.1% concentration (data not shown), were selected for solubilization assays. Following cell lysis and the STEAP1 recovery procedure, the resulting pellets were resuspended in 10 mM MES buffer at pH 6.2 with either 0.1% (*v*/*v*) of 5-Cyclohexyl-1-Pentyl-β-D-Maltoside (CYMAL-5), n-Decyl-β-D-Maltoside (DM), Nonidet P-40 (NP-40) or Genapol X-100 (GEN). In Figure 4A, it is observed that the Maltoside-based detergents were more effective in solubilizing STEAP1, with DM exhibiting the strongest band intensity. As for GEN and NP-40, very little difference can be observed from the control sample. In fact, it seems that GEN actually causes some degradation of STEAP1.

In the initial batch, a lysate dilution of 1:4 was used, as previously described [35]. However, since this lysate dilution, and therefore total initial protein concentration, resulted in the formation of large molecular weight bands exceeding 245 kDa, it was decided to assess if the total protein concentration was inducing aggregation events. So, a simplified batch, with only three steps consisting of binding (10 mM MES buffer at pH 6.2), washing (10 mM Tris pH 8) and elution (10 mM Tris pH 11) steps, was utilized to screen an array of initial lysate dilutions ranging from 1:4 to 1:20. Results are displayed in Figure 4B. Indeed, it seems that 1:4 dilution forms large molecular weight complexes and compromises analysis. Starting from 1:6 (total protein concentration of ~7 mg/mL), and moving forwards, some migration of STEAP1 to ~63 kDa was observed. Further, the bands at the top of the membranes remain present even at the most diluted samples of 1:20 (~2.15 total protein in mg/mL). This might indicate that STEAP1 aggregation is not the main driving force for the formation of these high molecular weight complexes. Nevertheless, considering the information of both screenings, from this point forward, all batches were performed with an initial lysate dilution of 1:6, since this dilution degree allows for the clarification of the highest amount of STEAP1 in each batch run. Also, 0.1% (*v*/*v*) DM was included in all buffers in order to solubilize and maintain STEAP1 stability throughout the batch runs. 

In fact, just applying these two optimized parameters in conjunction to the initial batch workflow with calcium-crosslinked GG microspheres, it was possible to bind the great majority of STEAP1 and start eluting it at pH 9.2, by charge neutralization (Figure 5). Moreover, up until this point, different microsphere volume ratios were tested. These were 20 mL and 35 mL of GG microspheres to 6 mL of buffer applied in each batch step. The 35 mL of GG microspheres exhibited better binding results for our membrane protein target and were selected for further analysis. The results from the ionic exchange strategy were very similar for nickel-crosslinked microspheres, in regard to elution profiles and protein content in each batch step (data not shown).

### 2.3. Batch Method for the Capture of STEAP1

#### 2.3.1. Affinity Strategy for STEAP1 Capture Using Nickel-Crosslinked GG Microspheres

Because the results from the ionic exchange strategy yielded equivalent results for both types of microspheres it would be redundant to develop the same approach for nickel- and calcium-crosslinked GG microspheres. Instead, nickel-crosslinked GG microspheres were used to capture recombinant STEAP1 through its 6xHis-Tag, simulating IMAC retention mechanisms, where elution would be prompted by increasing imidazole concentrations. Previous internal data had demonstrated that STEAP1 in a nickel IMAC column following an imidazole stepwise elution scheme (10 mM, 50 mM, 175 mM, 300 mM, 500 mM; elution profile adapted from [53]) started eluting at 175 mM imidazole (data not shown). Therefore, in the present work it was decided to set up a batch roadmap with fixed pH at 9.2 to eliminate or reduce any electrostatic interaction as much as possible. The latter optimized batch through lysate dilution, DM solubilization and microsphere ratio tuning exhibited positive results, yet a large degree of complexation was still present. To tackle this issue, a moderate amount of salt was also added to the buffers, to promote a slight salting-in effect and promote STEAP1 stabilization. Salt stabilization had been previously demonstrated for Rhodopsin, a structurally similar transmembrane protein [54]. The affinity batch was set up with a binding step consisting of 10 mM Tris at pH 9.2 with 150 mM NaCl and 5 mM imidazole and elution steps with the same amount of salt but with increasing concentrations of imidazole, corresponding to 175, 300 and 500 mM imidazole in 10 mM Tris pH 9.2. Following this gradient step mode, STEAP1 seemed to elute equally in all elution steps, suggesting that our target does in fact start eluting at 175 mM similar to IMAC. However, unlike IMAC, 175 mM imidazole was not enough to fully elute STEAP1 in a single step (data not shown). To address this elution profile, the batch was condensed to three steps, where binding would remain equal, followed by a washing step with 50 mM imidazole to remove any non-specific protein binding that may have occurred in the microspheres, and then a final elution step with 500 mM imidazole (the highest concentration from previous batch). 

Highlighted in Figure 6, with the described conditions, most of STEAP1 was captured through an affinity approach. However, it seems that over half was eluted in the supposed washing step with 50 mM imidazole. This indicates that a 3.5 times lower imidazole concentration can elute STEAP1 in a GG batch method as opposed to the necessary 175 mM in IMAC. Furthermore, a great deal of degradation (~17 kDa) was observed for the first time in all batch runs. Due to the degradation and the fact that it was not possible to recover the majority of STEAP1 in a single step, the samples recovered from nickel-crosslinked GG microspheres batches were excluded from further purification.

#### 2.3.2. Ionic Strategy for STEAP1 Capture Using Calcium-Crosslinked GG Microspheres

Similar to the affinity strategy, for the ionic batch strategy using calcium-crosslinked GG microspheres, salt was introduced to attain the same salting-in effect. However, instead of a fixed concentration of 150 mM, this assay explored the increase in intra-step NaCl levels in order to streamline the batch method to mimic a standard ionic exchange chromatography. In the optimization batch, it was noticed that although STEAP1 eluted mostly at pH 9.2, there was still a fraction of STEAP1 only being eluted at pH 11 buffer base. To recover as much target protein as possible in a single step, a switch to a single elution step and the replacement of the previous elution step at pH 9.2 with an additional washing step were made, in order to enhance the removal of impurities. In this manner, the ionic exchange batch remained a four-step batch with the conditions underlined in Figure 7A. In the binding step, practically all STEAP1 bound to calcium-crosslinked GG microspheres at pH 6.2 with no salt. Then, the minimal losses that were observed for the initial wash step at pH 8 with 100 mM NaCl, quickly turned into substantial losses by the increase to 200 mM NaCl at the same pH level. As intended, the bulk of STEAP1 was recovered in the final step by charge repulsion induced by the highest amount of salt.

Although a clearer sample was obtained in the end of the four-step batch system, a significant loss was observed in line III. To tackle this issue, the batch was condensed into three steps by removing the washing step with 200 mM NaCl and by concentrating this sample at pH 11 with 500 mM NaCl. However, when swapping to the condensed batch, mixed results were observed. As highlighted in Figure 7B, while all of STEAP1 was retained during the binding step and remained bound during the washing step, target samples returned to a fully complexed state in the elution step, even with all the optimizations previously described. Considering all the results thus far and GGs molecular weight of 500 kDa [55], we suspected that STEAP1 was forming complexes with the microspheres beyond simple ionic interactions. To assess the strength of this complexation, fully complexed samples were coupled with a co-immunoprecipitation polishing step. Indeed, it appears the antibody-STEAP1 interaction is stronger than GG-STEAP1, since post Co-IP STEAP1 was recovered in its monomeric form (Figure 8). SDS-PAGE of Co-IP STEAP1 shows a high degree of purity, although a major unidentified protein can be seen between ~63 and ~75 kDa. By analysis of the proteome from *Komagataella pastoris* X-33 reported by Huang and coworkers [56] and by taking into account proximity to the expected molecular weight and isoelectric point, we were able to identify Ferric and cupric reductase (FRE2) and Ferrioxamine B (SIT1) as possibilities for this unidentified protein. Summarizing the presented results, it appears that a simple GG batch can successfully act as a primary purification step. Even in the worst case scenario, where STEAP1 fully complexes with GG microspheres, a Co-IP polishing step can be applied to obtain a purified sample. Further, Co-IP seems to also be able to fix the aggregation issues derived from upstream stage (~63 kDa in lysate to ~35 kDa monomeric form).

## 3. Discussion

STEAP1 has been appointed as a putative biomarker and therapeutic agent in a plenitude of cancers, with higher expression levels in PCa. Unlike other STEAP family members (2–4), STEAP1 does not contain an N-terminal NADPH binding domain. Instead, it has a long tail that sits in the intracellular domain with undiscovered functions. Further, STEAP1 has been predicted to play a role in a slew of signaling pathways [57,58,59,60] and we speculate this STEAP1 disposition could be related to signal transduction. As previously mentioned, it is imperative to uncover the full STEAP1 structure, in order to explore the role this protein plays in biological systems. For this, improvements to current bioprocesses are necessary. Earlier, our research group made strides in optimizing the production of recombinant human STEAP1 in a mini-bioreactor platform [61]. The produced recombinant protein fractions increased the proliferation of prostate cancer cell lines, indicating they acquire active conformation [61]. Here, we tackled the downstream portion in an attempt to improve purification yields, resorting to a simple batch method with GG microspheres. Both calcium- and nickel-crosslinked GG microspheres were produced through a previously optimized W/O emulsion method, resulting in average diameters of 330.37 ± 11.38 µm and 239.06 ± 5.43 µm, respectively. These values are lower than those reported for GG microspheres produced by ionotropic gelation [45,62]. In fact, these values are approximately two-fold lower than those reported by Narkar and coworkers, using similar methodology and processing parameters [62]. Indeed, as the size of microspheres decreases, it is expected an improvement in specific surface area, and consequently an improvement in adsorption capacity [63]. However, no evident change was observed between calcium and nickel microspheres after the ionic optimization batch (data not shown). 

The initial batch strategy was based on a very simple ionic interaction. In the ionic approach, GG would always present a negative charge (pKa = 3.5; [64]) and STEAP1 would bind at pH 6.2 with a positive charge, and be eluted at >pH 9.2, by electrostatic repulsion. However, in the initial batch, only large molecular weight bands were observed in the immunoreactive assays. At first, it was suspected that STEAP1 might be getting stuck inside or in certain cavities in GG microspheres. However, calcium-crosslinked GG microspheres SEM images present a mostly smooth surface with no apparent pores or cavities. Next, it was suspected that STEAP1 was forming large aggregates. To improve STEAP1 stability during the batch, several optimizations were initiated. First, a series of non-ionic detergents were applied in the solubilization of STEAP1. These detergents were selected mainly because they are non-denaturant and can ensure the biological function of membrane proteins, as opposed to ionic and zwitterionic detergents which are harsher and often lead to deactivation or denaturation of membrane proteins [51,65]. From the selected detergents, DM presented the best solubilization potential. Similar to most membrane proteins, STEAP1 is best solubilized by alkyl maltopyranosides. Indeed, approximately 50% of membrane proteins in the Membrane Proteins of Known 3D Structure database were solubilized by alkyl maltopyranoside detergents both in the purification and crystallization phases of structure determination [66]. 

Thereafter, the influence of total protein content in the batch was assessed by screening an assortment of different dilutions from 1:4 to 1:20. Starting from a dilution of 1:6 (~7 mg/mL) and moving forwards, some migration was observed to ~63 kDa. This could suggest that aggregation was in fact a contributing factor; however, even at dilutions of 1:20 (~2.15 mg/mL) the presence of large weight aggregates was constant throughout immunoreactive assays. When applying both DM and 1:6 dilution in the optimized batch (Figure 5), it was possible to minimize the large molecular weight complexes. Furthermore, the remaining complexes were mainly localized at the elution step with pH 9.2, where the majority of STEAP1 eluted. When the SDS-PAGE of all assays is compared, it is clear that all lanes are very similar in protein composition. So, if hetero-aggregation was the cause, it would be expected that other proteins would interfere with antibody detection of STEAP1. Yet, the intensity of the complexes band seems to mimic the intensity to which STEAP1 is present in each batch step. In addition, the presence of small STEAP1 aggregates (~48 and ~63 kDa) is not considered a concern, since they have been previously associated with the recombinant production steps [61]. In fact, Kim and coworkers ventured as far as to call them the dimeric and trimeric STEAP1 [9]. 

In order to further reduce the occurrence of complexation, the addition of salt was carried out to promote a small salting-in effect and stabilize STEAP1. In the four-step batch (Figure 7A), only a very modest benefit was observed in the reduction of complexes. Later, when condensing to a three step batch to minimize STEAP1 losses, mixed results were observed. Indeed, STEAP1 was fully collected in the intended elution step without losses, but it came fully complexed. During the intra-step centrifugation steps and posterior supernatant collection, it was observed that some GG microspheres did not sediment completely and were recovered in the supernatant. Even with several optimization studies onto this capture method, it was not possible to identify a schema that could fully separate the GG microspheres from the target protein, without substantially compromising target protein yields and batch reproducibility. So, we suspect that GG with 500 kDa was forming complexes with STEAP1 and increasing amounts of microspheres recovered would prompt higher rates of complexation. Indeed, these complexes seem to originate outside the scope of standard ionic interactions since electrostatic repulsion and 500 mM NaCl should have been more than enough to disrupt these interactions. STEAP1 has been previously predicted to act both as an ionic channel and to modulate the concentration of small ions, calcium included [6,67]. Perhaps STEAP1 functions as a calcium transmembrane channel and forms GG-STEAP1 complexes by the mediation of latent affinity towards the calcium crosslinker. Further, nickel cellular uptake has been shown to be calcium dependent, with some evidence suggesting that it crosses the plasma membrane through calcium channels [68], which might suggest why nickel-crosslinked GG microspheres suffered from the same complex formation. This phenomenon highlights the effect that inherent biomolecule properties can have on establishing biointeractions with GG microspheres. For instance, our research group has previously relied on similar GG microsphere batch methods as the primary purification technique for the purification of soluble proteins [35] and plasmid DNA [36], without the rise of any complexation issues.

Nevertheless, the sample recovered from the condensed ionic strategy batch was coupled with a Co-IP polishing step, since this technique is highly specific and selective in the detection of physical protein interactions [69]. Results indicated that the formation of the antibody-STEAP1 immunoconjugates was stronger than the affinity between GG-STEAP1 complexes, as the latter complexes were disrupted and STEAP1 was recovered in its purified monomeric form. Further, our Co-IP results for recombinant human STEAP1 were very similar to those reported by a hydrophobic interaction chromatographic (HIC) step coupled with Co-IP purification workflow, with lysates from LNCaP cells [69]. However, when assessing the results reported by Oosterheert and coworkers [10], the sequential chromatographic techniques employed, especially considering the stronger and more selective affinity explored, yields samples with a substantially higher level of purity. On the other hand, it is unclear how this workflow would react to the lysate concentration and complexity level reported here. Our research group has recently reported the purification of recombinant STEAP1 from mini-bioreactor cultures by coupling either an HIC or IMAC primary purification step with a Q-Sepharose anion exchanger polishing step, yielding comparable purification results, by SDS-PAGE analysis [70]. Unfortunately, considering the inability to completely separate GG from STEAP1, the presence of glucose moieties in the GG backbone and the fact that reducing sugars are a known interference in the BCA protein quantification assay [71], it was not possible to quantify the recovered samples, and therefore provide quantitative comparisons to current report methods. Furthermore, STEAP1 needs to form heterotrimeric ensembles with other STEAP members to attain metal reductase activity [10], making quantification via an enzymatic iron reduction approach inaccessible, since recombinant STEAP1 was used in the batch method. Indeed, new approaches that allow the quantification of the recovered samples should be addressed in future research.

## 4. Materials and Methods

### 4.1. Materials

Ultrapure reagent-grade water was obtained from a Milli-Q system from Millipore/Waters. Gellan Gum (Gelzan^™^, Gelrite^®^), glass beads, lysozyme, deoxyribonuclease I (DNase), bromophenol blue, MES hydrate and MES sodium salt were acquired from Sigma-Aldrich Co. (St. Louis, MO, USA). Tris-base, tween-20, glycine, imidazole, sodium chloride (NaCl), nickel chloride hexahydrate (NiCl_2_.6H_2_O) and methanol were purchased from ThermoFischer Scientific (Waltham, MA, USA). Calcium Chloride dihydrate (CaCl_2_.2H_2_O) and sodium dodecyl sulfate (SDS) were obtained from PanReac Applichem (Darmstadt, Germany). β-mercaptoethanol and N,N,N′,N′-Tetramethylethylenediamine (TEMED) were acquired from Merck (Darmstadt, Germany). Bis-Acrylamide/Acrylamide 40% and NZYColour Protein Marker II were obtained from GRiSP Research Solutions (Oporto, Portugal) and NZYTech (Lisbon, Portugal), respectively. All other reagents and supplies were of analytical grade.

### 4.2. Gellan Microspheres Production

GG microspheres were produced through a water-in-oil (W/O) emulsion method, previously optimized by our research group through a design of experiments approach [72]. Briefly, we dissolved a 1.41% GG solution at 90 °C and 300 rpm for 15 min. Then, the GG solution was extruded drop by drop from a syringe with a 21G needle attached to a syringe pump (Harvard Apparatus, Cambourne, UK). The flow rate was set to 75 µL/min and the solution was dripped from a height of approximately 20 cm into 100% vegetable cooking oil previously heated to 100 °C under strong agitation. Next, the microspheres were reinforced with either Ca^2+^ or Ni^2+^, by the addition of a 200 mM crosslinker solution to the emulsions at 750 rpm and room temperature during 30 min. Subsequently, excess oil was removed with 70% ethanol in a vacuum filtration system with 11 µm pore size filter paper (VWR, Radnor, PA, USA). Finally, GG microspheres were washed with water and stored in 10 mM MES buffer pH 6.2, at 4 °C, until they were used in capture trials.

### 4.3. Gellan Microspheres Characterization

The produced GG microspheres were characterized following the procedures described by Gomes and coworkers [35].

#### 4.3.1. Semi-Optical Microscopy

First, the average diameter of GG microspheres was assessed through semi-optical microscopy. So, microspheres were seated into microscope slides and visualized at 10× magnification. Six different images (n = 6) with a total of 46 measurements were obtained and the mean diameter was assessed.

#### 4.3.2. Scanning Electron Microscopy (SEM)

Surface morphology of GG microspheres was evaluated through SEM, using a Hitachi S-3400 N microscope (Tokyo, Japan). The microspheres were distributed onto an aluminum support with a carbon base and frozen at −20 °C. Then, several representative images were obtained using different magnifications, with a backscattered electron (BSE) 3D detector.

#### 4.3.3. Elemental Analysis and Chemical Composition

To shed light on the elemental composition of produced microspheres, and confirm the incorporation of calcium, energy dispersive X-ray spectroscopy (EDX) was conducted. Still frozen, post SEM snapshot acquisitions, microspheres were analyzed via a QUANTAX 400 detector (Bruker, Billerica, MA, USA).

#### 4.3.4. Fourier-Transformed Infrared Spectroscopy (FTIR)

FTIR was utilized to gauge the appropriate formation of GG microspheres and to ensure divalent ion crosslinking. For this, samples were lyophilized, and spectra were acquired using an FTIR spectrophotometer (Nicolet iS10; ThermoFischer Scientific, Waltham, MA, USA) for both GG powder and GG microspheres. The equipment was managed in the OMNIC Spectra software (ThermoFischer Scientific), and spectra were collected operating in ATR mode with an average of 120 scans on wavenumbers ranging from 400–4000 cm^−1^, at a resolution of 32 cm^−1^. 

### 4.4. Mini-Bioreactor Production and Recovery of STEAP1

The production and lysis of recombinant STEAP1 was performed as described by Duarte and coworkers [61]. Briefly, *Komagataella pastoris* X-33 Mut^+^ were selected on yeast peptone dextrose (YPD) plates, at 30 °C. Then, a single colony was chosen and transferred to shaker flasks with 100 mL of Buffered minimal glycerol medium (BMGH) and grown overnight at 30 °C and 250 rpm until OD_600nm_ reached a value between 5–6. Then, an appropriate volume was collected so that the initial fermentation OD600_nm_ was equivalent to 0.5 and was deposited into 750 mL vessels from a mini bioreactor platform with 250 mL of basal salt medium (BSM) supplemented with Zeocin^™^ and a trace metal solution (SMT). The STEAP1 biosynthesis was divided into 3 main stages. First, a standard batch occurred until depletion of glycerol, detected by a sharp increase in dissolved oxygen. After, a glycerol fed-batch phase was carried out for 2 h, in order to improve biomass levels, followed by a 1 h transition phase where methanol was introduced to the feed to prepare the culture for a new carbon source. The third stage consists of a methanol feed strategy inducing the AOX promoter in the cells and stimulating the expression of His-tagged recombinant human STEAP1 (rhSTEAP1). Finally, the cells were retrieved by centrifugation for 10 min at 1500× *g* and 4 °C.

To recover STEAP1, *K. Pastoris* cells were resuspended in lysis buffer (50 mM Tris, 150 mM NaCl, pH 7.8) supplemented with a protease inhibitor cocktail (Hoffmann-La Roche, Basel, Switzerland). Lysozyme (1 mg/mL) was added to the mix at room temperature for 15 min. After enzymatic digestion, the mixture was transferred to a falcon with glass beads in a ratio of 1:2:2, respectively, 1 g biomass, 2 mL of lysis buffer and 2 g of beads. Subsequently, mechanical lysis was executed through seven vortex cycles, interposed by 1 min intervals on ice. Next, the cell fragments and glass beads were separated by a 5 min 500× *g* centrifugation at 4 °C, forming a three layer system, consisting of supernatant, pellet and glass beads. The supernatant was discarded and the pellet was resuspended in lysis buffer. The glass beads are naturally separated from the solution by differential density values, allowing recovery of the resuspended pellet. In turn, the recovered solution was supplemented with DNase (1 mg/mL) and centrifuged at 16,000× *g* for 30 min at 4 °C. The supernatant was discarded, and the pellet was resuspended in the appropriate binding buffer for the batch method capture step. The total protein content in the lysates was quantified by the Pierce BCA Protein Assay Kit (ThermoFischer Scientific) following the manufacturer’s instructions.

### 4.5. Batch Method for the STEAP1 Capture

The employed batch method was adapted from the batch described by Gomes and coworkers for the capture of COMT [35]. First, GG microspheres were equilibrated with an appropriate buffer for the capture step. Then, the batch consisted of three main stages: binding, washing and elution. The binding or capture step was initiated by the addition of the lysate in an appropriate dilution to the microspheres. This step was carried out for a total of 4 h, at 4 °C under gentle tube agitation. This was followed by a centrifugation at 500× *g* for 8 min and recovery of the supernatant, corresponding to the protein fractions that did not bind to the microspheres. The washing and elution steps follow the same profile with adequate buffers during 1 h, yielding the eluted fractions. The batch was applied to GG microspheres crosslinked with calcium or nickel ions. For calcium, an ionic exchange strategy was chosen, by manipulation of pH, ranging from 6.2 to 11, and ionic strength, by manipulation of NaCl concentrations ranging from 0 to 500 mM, in order to recover STEAP1. For nickel, an affinity method similar to immobilized metal affinity chromatography was used, where STEAP1 was bound to the microspheres through its His-Tag and eluted by varying imidazole concentrations, ranging from 5 to 500 mM in total concentration. The recovered fractions were concentrated and desalted with Vivaspin concentrators (10,000 MWCO) and stored at 4°C until further purity or immunoreactivity analysis.

### 4.6. Co-Immunoprecipitation

The clarified sample from the batch method was coupled with a final polishing co-immunoprecipitation (Co-IP) step. Co-IP was performed following the manufacturer’s protocol for Protein A/G PLUS-Agarose Immunoprecipitation Reagent (sc-2003, Santa Cruz Biotechnology, Dallas, TX, USA) with slight modifications. Succinctly, STEAP1 clarified samples were incubated for 1 h at 4°C with anti-STEAP1 mouse monoclonal antibody (B-4, Santa Cruz Biotechnology, Dallas, TX, USA), followed by overnight incubation with agarose beads with constant stirring. Conjugated complexes were recuperated by centrifugation at 1000× *g* for 5 min at 4 °C. Supernatant was discarded, the complexes were washed with PBS and then resuspended in electrophoresis loading buffer (refer to Section 4.7). The agarose beads were separated from the antibody-STEAP1 complexes by the combinatory effect of sample boiling at 100 °C and 5% (*v*/*v*) β-mercaptoethanol.

### 4.7. SDS-PAGE and Western Blot

Reducing SDS-polyacrylamide gel electrophoresis (SDS-PAGE) was performed according to the Laemmli method [73]. In essence, samples from the batch method were boiled for 5 min at 100 °C and resolved in two 12.5% SDS-PAGE gels at 120 V. Then, one gel was stained by Coomassie blue solution, while the other was transferred into a PVDF membrane (GE Healthcare, Wauwatosa, WI, USA) at 750 mA for 90 min at 4 °C. The membranes were blocked in 5% non-fat milk and incubated overnight with anti-STEAP1 mouse monoclonal antibody 1:300. Afterwards, following a 2 h incubation with goat anti-mouse IgG-HRP 1:5000 (sc-2005, Santa Cruz Biotechnology, Dallas, TX, USA), STEAP1 immunoreactivity was analyzed with ChemiDoc™ MP Imaging System after incubation with ECL substrate (Bio-Rad, Hercules, CA, USA).

## 5. Conclusions

In summary, we developed a simple batch method using GG microspheres for the capture of STEAP1 from mini-bioreactor *Komagataella pastoris* lysates, exploiting affinity and ionic interactions. The affinity strategy using nickel-crosslinked GG microspheres proved to be highly specific, binding most of STEAP1 through the nickel-histidine affinity interaction. Yet, recovering the bound target in a single step was challenging, since the formed interaction was highly sensitive, even to mild imidazole concentrations. On the other hand, the ionic batch using calcium-crosslinked GG microspheres is a more robust method, capable of yielding nearly all of STEAP1 in a single step, albeit often in a complexed state. This complexation seems to be STEAP1-specific, and the chemical nature of such strong complexes should be addressed in future research. Nevertheless, coupling a Co-IP polishing step to the batch method yields STEAP1 with a high degree of purity and completes the purification workflow. Still, it is relevant in future work to conduct structural and functional biochemical assays to ensure the recovered monomeric STEAP1 retains appropriate secondary structure and typical structural features, after being subjected to the batch method and Co-IP workflow. 

Overall, the ionic batch method is simple, fast, cost-effective and can be applied as a primary capture step for STEAP1. Further, since the average isoelectric point of membrane proteins seems to be between 8.5 to 9.0 [74], it is safe to assume that this simple ionic GG batch method can be extended to the capture of other relevant membrane proteins with clinical interest.

## Figures and Tables

**Figure 1 ijms-24-01949-f001:**
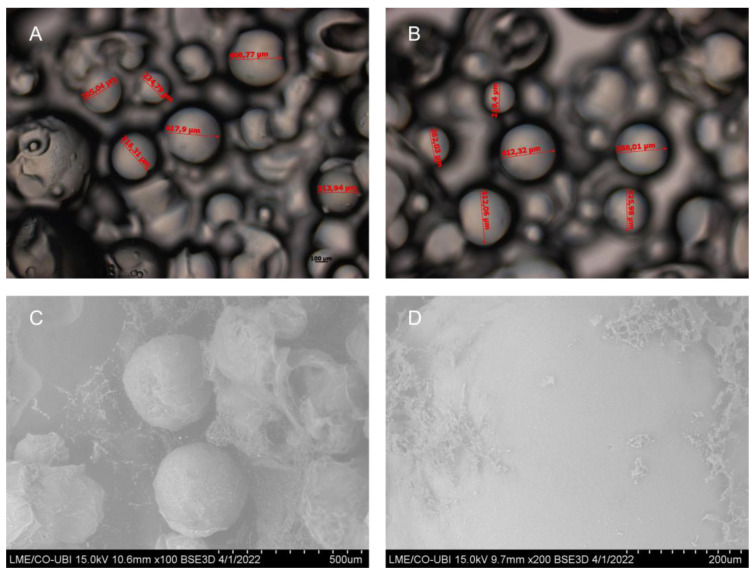
Semi-optical snapshots of calcium-crosslinked GG microspheres (**A**,**B**) and SEM images at ×100 magnification (**C**) and ×200 magnification (**D**).

**Figure 2 ijms-24-01949-f002:**
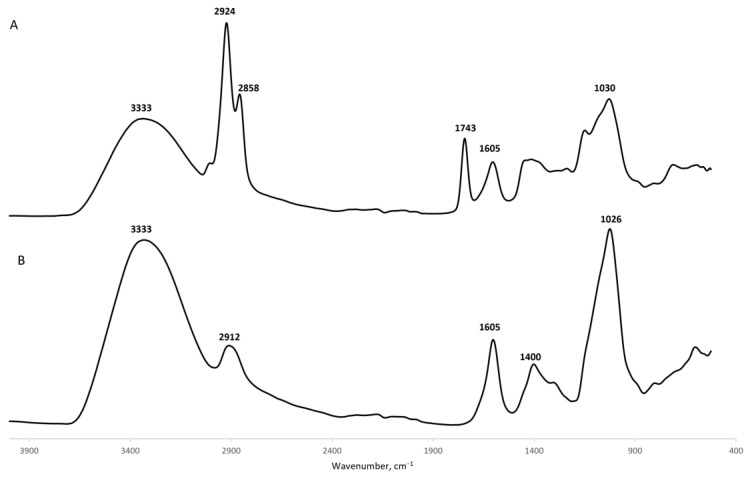
FTIR spectra (absorbance vs. wavenumber) of calcium-crosslinked GG microspheres (**A**) and GG powder (**B**).

**Figure 3 ijms-24-01949-f003:**
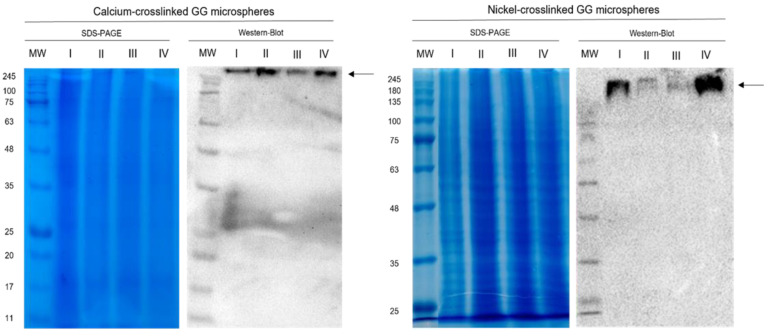
SDS-PAGE and Western blot of the recovered supernatants from the initial batch for both calcium- and nickel-crosslinked microspheres (35 mL GG microspheres for both ions represented). MW—molecular weight; I—sample that did not bind to GG microspheres at 10 mM MES pH 6.2; II—washing step with 10 mM Tris pH 8; III—elution step with 10 mM Tris pH 9.2; IV—elution step with 10 mM Tris pH 11; arrows indicate STEAP1 complexes.

**Figure 4 ijms-24-01949-f004:**
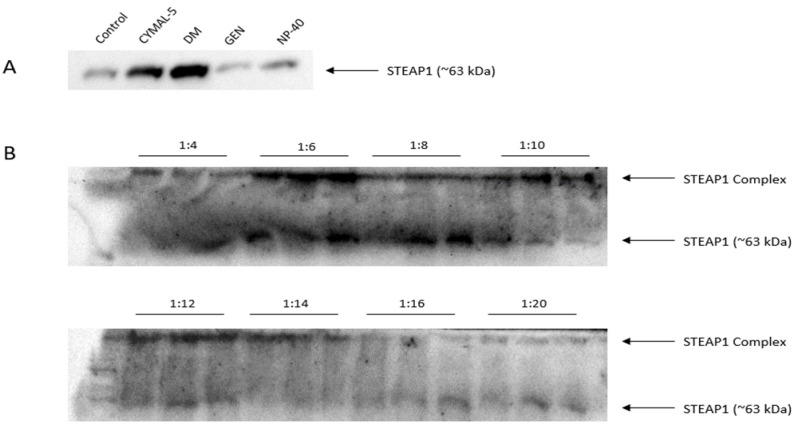
Western blot of the detergent screening for solubilization of STEAP1 (control represents insolubilized lysate samples) (**A**) and recovered supernatants from the initial batch lysate dilution screening following a simple three step sequence per dilution: binding—10 mM MES pH 6.2; washing—10 mM Tris pH 8; elution—10 mM Tris pH 11 (**B**).

**Figure 5 ijms-24-01949-f005:**
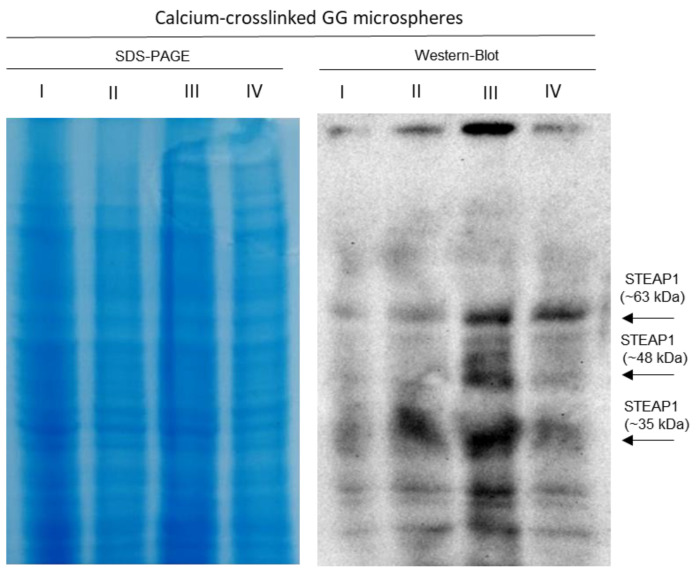
SDS-PAGE and Western blot of the recovered supernatants from the optimized batch for calcium-crosslinked GG microspheres (35 mL GG microspheres); I—sample that did not bind to GG microspheres at 10 mM MES pH 6.2; II—washing step with 10 mM Tris pH 8; III—elution step with 10 mM Tris pH 9.2; IV—elution step with 10 mM Tris pH 11.

**Figure 6 ijms-24-01949-f006:**
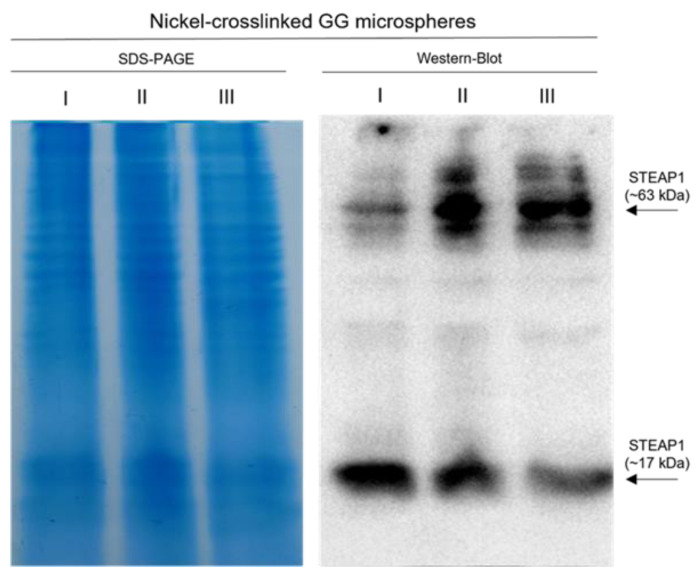
SDS-PAGE and Western blot of the recovered supernatants from affinity batch with nickel-crosslinked GG microspheres; I—sample that did not bind to GG microspheres at 10 mM Tris pH 9.2 with 150 mM NaCl and 5 mM imidazole; II—washing step with 10 mM Tris pH 9.2 with 150 mM NaCl and 50 mM imidazole; III—elution step with 10 mM Tris pH 9.2 with 150 mM NaCl and 500 mM imidazole.

**Figure 7 ijms-24-01949-f007:**
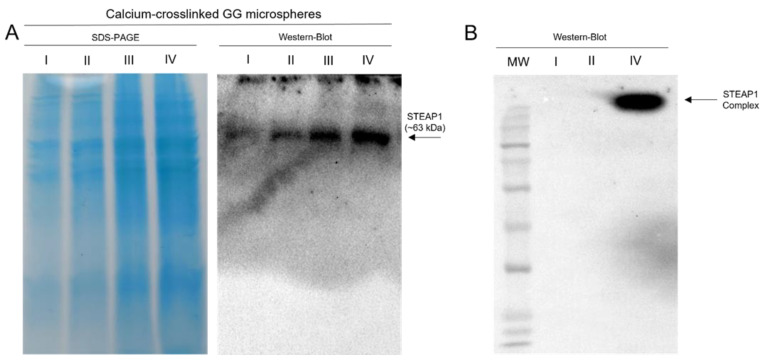
SDS-PAGE and Western blot of the recovered supernatants from the ionic exchange batch for calcium-crosslinked GG microspheres for both a four-step batch (**A**) and condensed batch (**B**) (35 mL GG microspheres); I—sample that did not bind to GG microspheres at 10 mM MES pH 6.2; II—wash step with 10 mM Tris pH 8 and 100 mM NaCl; III—elution step with 10 mM Tris pH 8 and 200 mM NaCl; IV—elution step with 10 mM Tris pH 11 and 500 mM NaCl.

**Figure 8 ijms-24-01949-f008:**
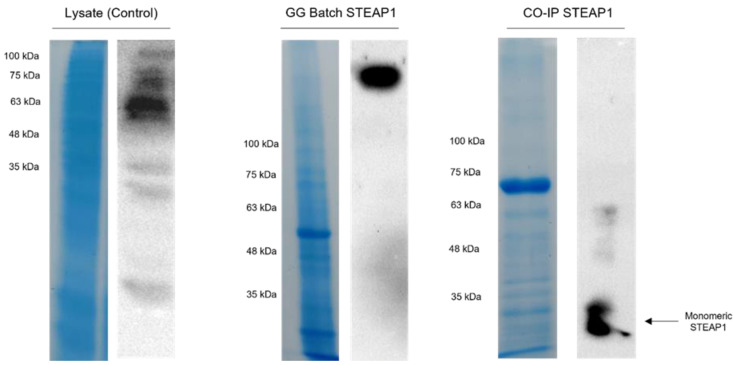
SDS-PAGE and Western blot of the entire purification workflow: the initial total protein content in *K.pastoris* lysate, the clarified sample from calcium-crosslinked GG batch and the purified co-immunoprecipitated STEAP1.

**Table 1 ijms-24-01949-t001:** Elemental composition of GG microspheres through EDX.

Element	Calcium-Crosslinked GG Microspheres		Nickel-Crosslinked GG Microspheres	
	C Norm. [wt%]	C Atom. [at%]	C Norm. [wt%]	C Atom. [at%]
Carbon	31.87	38.72	39.19	47.13
Oxygen	66.54	60.70	57.73	52.12
Calcium	1.59	0.58	-	-
Nickel	-	-	3.08	0.76
Total	100.0	100.0	100.0	100.0
Ref.	-		[35]	

## Data Availability

Not applicable.

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
