# Peer review of "Specific Six-Transmembrane Epithelial Antigen of the Prostate 1 Capture with Gellan Gum Microspheres: Design, Optimization and Integration"

_ijms, 2023, doi:10.3390/ijms24031949_

Round 1
Reviewer 1 Report
This manuscript describes the use of calcium crosslinked and nickel crosslinked gellan gum microspheres for purification of recombinant STEAP1 protein from bacterial lysate. The work is interesting and well written, but there are a few issues which should be addressed before publication.
In the microsphere production details, the authors indicate that the microspheres were “dried with water” (line 468). It is not clear, exactly what was done in this step.
In the protein recovery process, the authors state (Lines 518-520): “Next, the cell fragments and glass beads were separated by a 5 min 500g centrifugation at 4â—¦C. The pellet was resuspended in lysis buffer supplemented with DNase (1 mg/mL) and centrifuged at 16000g for 30 min at 4â—¦C.” One would expect that the glass beads would be incorporated into the pellet by the centrifugation step. If the pellet was then resuspended, the beads would still be present. There is no step further along in which the glass beads are removed. The authors have to clarify the procedure to indicate what happens to the glass beads in the procedure.
Likewise, after the protein incubation step in Lines 532-533, the authors state: “This was followed by a centrifugation at 500g for 8 min and recovery of the supernatant.” One would expect the microspheres to which the protein bound would be in the pellet after the centrifugation. Why was the supernatant collected? If this was collected simply to measure what was not bound to the GG spheres, that is fine, but the way it is written suggests that the supernatant is carried to the next step in the process, which does not make sense.
In Line 188, it indicates that the recovered protein had an “elevated molecular weight of >245 kDa.” It does not state previously what MW is actually expected for the recombinant protein being produced for this work. This information should be included here so that the reader can immediately understand what MW band they should be expecting in the gel and blot lanes in the figures.
For Section 2.3.1, the authors have to make it clear that the STEAP1 protein they are capturing is the recombinant version that has the His-Tag, and should identify it as such (rhSTEAP1),so that the reader does not think that the native version could be purified in this manner.
In lines 359-361, the authors acknowledge that smaller gellan beads would improve absorption capacity. The work by Moslemy (Enzyme Microb. Technol. 30 (2002) 10) showed production of calcium crosslinked gellan gum beads from 12 to 135 uM in diameter using the same type of water in oil emulsion process. Since these spheres are significantly smaller than what was produced here, some justification for the use of the significantly larger beads should be provided.
Line 375: “do” should be “to”
It was observed (lines 401-403) that some of the microspheres were not sedimented during the centrifugation step to collect the wash and elution fractions. If this was the case, why did the authors not simply increase the speed or time of the centrifugation step to address this issue?
Line 408: “origin” should be “originate”
Author Response
Reviewer 1 - Overall assessment of the Revised manuscript.
This manuscript describes the use of calcium crosslinked and nickel crosslinked gellan gum microspheres for purification of recombinant STEAP1 protein from bacterial lysate. The work is interesting and well written, but there are a few issues which should be addressed before publication.
Comment 1- “In the microsphere production details, the authors indicate that the microspheres were “dried with water” (line 468). It is not clear, exactly what was done in this step.”
Authors reply: We appreciate the comment made by the Reviewer. This step occurs in the vacuum filtration system, following the removal of oil by 70% ethanol. Indeed, the water is utilized to remove any remnants of ethanol or oil in the microspheres, while the vacuum pump aspirating the liquid is responsible for the “drying” process. The manuscript was altered to “washed with water” to better reflect the procedure, please see line 491 in page 14.
Comment 2- “In the protein recovery process, the authors state (Lines 518-520): “Next, the cell fragments and glass beads were separated by a 5 min 500g centrifugation at 4â—¦C. The pellet was resuspended in lysis buffer supplemented with DNase (1 mg/mL) and centrifuged at 16000g for 30 min at 4â—¦C.” One would expect that the glass beads would be incorporated into the pellet by the centrifugation step. If the pellet was then resuspended, the beads would still be present. There is no step further along in which the glass beads are removed. The authors have to clarify the procedure to indicate what happens to the glass beads in the procedure.”
Authors reply: The authors acknowledge the Reviewer for this comment and the possibility to clarify the manuscript. From the first 5 min 500g centrifugation, we retrieve a sample with three layers, consisting of supernatant, pellet and glass beads, and not a mixture of pellet and glass beads. The supernatant is discarded, the pellet is resuspended in lysis buffer and vortexed with the glass beads. However, given a few seconds, the glass beads deposit in the bottom of the falcon tubes and out of solution, by difference in density values, allowing us to pipette the sample and remove the glass beads. This information was integrated in the manuscript. Please see page 15, lines 543-547.
Comment 3- “Likewise, after the protein incubation step in Lines 532-533, the authors state: “This was followed by a centrifugation at 500g for 8 min and recovery of the supernatant.” One would expect the microspheres to which the protein bound would be in the pellet after the centrifugation. Why was the supernatant collected? If this was collected simply to measure what was not bound to the GG spheres, that is fine, but the way it is written suggests that the supernatant is carried to the next step in the process, which does not make sense.”
Authors reply: The authors acknowledge the Reviewer for this comment, which help us to significantly improve the manuscript. Indeed, the reviewer is correct, this sample was collected to measure the protein fraction that did not bind to the microspheres. The core principle and steps involved in the batch method are very similar to traditional chromatography. Briefly, the GG microspheres undergo an equilibration step with the buffer to set the ideal conditions for the binding step. Then, 6 mL lysate diluted in an appropriate concentration is added to the particulate matter and incubated for 4h. The aforementioned centrifugation step at 500g 8 min, separates the 35 mL GG microspheres and bound protein fraction from the non-bound fraction. The supernatant recovered corresponds to this non-bound fraction. However, please note that the centrifugation steps do not result in pelleted matter. The velocity and time were carefully selected to separate the solid matter from the liquid fraction, keeping the protein-microsphere bonds/interactions intact, similar to a chromatographic resin. Then, 6 mL of buffer with a given composition is added for 1h to promote the following washing/elution steps, with the following centrifugations yielding the eluted fractions. The manuscript was altered to clarify this issue. Please see page 15, lines 560-563.
Comment 4- In Line 188, it indicates that the recovered protein had an “elevated molecular weight of >245 kDa.” It does not state previously what MW is actually expected for the recombinant protein being produced for this work. This information should be included here so that the reader can immediately understand what MW band they should be expecting in the gel and blot lanes in the figures.
Authors reply: The authors acknowledge the Reviewer’s comment. Indeed, the molecular weight of the recombinant STEAP1 produced by Komagataella pastoris in mini-bioreactor cultures is expected in ~35, ~48 or ~63 kDa, as previously demonstrated [1]. The required information has been added to the manuscript. Please see section 2.2, pages 5 and 6, lines 185-187 and 190-191.
[1] Duarte, D.R.; Barroca-Ferreira, J.; Gonçalves, A.M.; Santos, F.M.; Rocha, S.M.; Pedro, A.Q.; Maia, C.J.; Passarinha, L.A. Impact of Glycerol Feeding Profiles on STEAP1 Biosynthesis by Komagataella Pastoris Using a Methanol-Inducible Promoter. Appl. Microbiol. Biotechnol. 2021, 105, 4635–4648, doi:10.1007/s00253-021-11367-y.
Comment 5- For Section 2.3.1, the authors have to make it clear that the STEAP1 protein they are capturing is the recombinant version that has the His-Tag, and should identify it as such (rhSTEAP1), so that the reader does not think that the native version could be purified in this manner.
Authors reply: The authors acknowledge the reviewer’s comment and the possibility to improve the manuscript. Indeed, it had been previously stated in the last paragraph on the Introduction section (lines 107-108) that this work is focused on recombinant STEAP1. However, to erase any possible confusion we have added “recombinant” in section 2.3.1 (“… were used to capture recombinant STEAP1 through its 6x His-Tag …”; page 8, line 257). This aspect is further reinforced in the discussion section and in the experimental section (section 5.4), making it clear we are dealing with recombinant STEAP1.
Comment 6- In lines 359-361, the authors acknowledge that smaller gellan beads would improve absorption capacity. The work by Moslemy (Enzyme Microb. Technol. 30 (2002) 10) showed production of calcium crosslinked gellan gum beads from 12 to 135 uM in diameter using the same type of water in oil emulsion process. Since these spheres are significantly smaller than what was produced here, some justification for the use of the significantly larger beads should be provided.
Authors reply: The authors acknowledge the reviewer’s question and the opportunity to clarify. When producing microspheres of any kind, two of the most critical aspects when controlling in regard to size, are polymer percentage and stirring speed. Indeed, Moslemy and coworkers proceeded with an emulsification method like ours. However, certain differences can be noted. First, in our method we carried out extrusion of the GG solution onto the vegetable oil solution, which is not the case in the proposed article. Next, we utilized a 1.41% gellan gum solution, nearly two-fold the concentration utilized by Moslemy. Indeed, the Reviewer is correct, as the higher viscosity, especially considering the extrusion process would result in higher overall average size of GG microspheres. However, we could argue that the third parameter, the stirring speed, is far more influential in this regard, as the authors of the proposed article utilized ranges from 1000-5500 rpm, as opposed to the fixed amount of 750 rpm. Indeed, with increases in stirring speed, it is expected decreases in microsphere size. Moslemy and team utilized a “quarter-circular paddle impeller assembled with a T-Line laboratory stirrer” to achieve such high rotational speeds. Moreover, when analyzing Table 2 of the proposed article, it is clear that the stirring speed has a large degree of effect on the final particle size, with increases in stirring speed leading to sharp decreases in final microsphere size. Unfortunately, we do not have access to this equipment or similar machinery. The GG microsphere production methodology described in the manuscript was optimized by our research group, through a Design of Experiment (DoE) approach, leading to the minimization of GG microsphere size, considering the equipment accessible to our research group [1]. Moreover, if the reviewer considers the methodology described by Narkar and coworkers (manuscript reference [62]) for production of calcium-crosslinked GG beads, where similar procedures and equipment were used, we can quickly see the merit in our production methodology in reducing microsphere size. Narkar and coworkers also proceed with an extrusion approach for formulation of GG microspheres with a more comparable viscosity (1.5% vs our 1.41% solution). However, with these conditions they obtained GG beads with over 2 times the final diameter sizes. This fact was reinforced in page 11, lines 360-362.
Coelho, J.; Eusébio, D.; Gomes, D.; Frias, F.; Passarinha, L.A.; Sousa, Â. Biosynthesis and Isolation of Gellan Polysaccharide to Formulate Microspheres for Protein Capture. Carbohydr. Polym. 2019, 220, 236–246, doi:10.1016/j.carbpol.2019.05.011.
Comment 7- Line 375: “do” should be “to”
Authors reply: This comment was properly addressed in the manuscript. Please see page 12, line 379.
Comment 8- It was observed (lines 401-403) that some of the microspheres were not sedimented during the centrifugation step to collect the wash and elution fractions. If this was the case, why did the authors not simply increase the speed or time of the centrifugation step to address this issue?
Authors reply: The authors acknowledge the Reviewer for this comment and the possibility to clarify the manuscript. Indeed, we attempted increases both in centrifugation speed and time to tackle this issue. For instance, we utilized 500 g 8 min, as opposed to the batch method this work was based on, which utilized 100 g 3 min (manuscript reference [35]). Further increases in centrifugation speed resulted in substantial protein precipitation, that would not be easily resolubilized with the addition of the next batch step’s buffer, in large part due to the microsphere:buffer ratios utilized (35 mL GG microspheres to 6 mL buffer). This means that we would be losing these protein fractions for the remainder of the assay, compromising total process yield and also giving rise to false positive results. For instance, protein fractions that would normally be eluted in a given buffer composition, might never be recovered due to excessive centrifugation force and impact batch process results and reproducibility. The aforementioned conditions, although not fully separating GG microspheres, minimized protein losses, and therefore, maximized total process yield. We would also like to reiterate that the protein fraction is meant to remain bound during the centrifugation steps, mimicking a chromatographic resin. Extensive centrifugation speeds would break these interactions, which as described would be hard to resolubilize. Furthermore, even if we were to explore this option of completely pelleting the protein fractions, we would have to remove the microspheres in each centrifugation step to be able to recover the pellet and then reintroduce them for subsequent batch steps, which would be inefficient. This issue was reinforced in the discussion (please see page 12, lines 406-411) :
“During the intra-step centrifugation steps and posterior supernatant collection, it was observed that some GG microspheres did not sediment completely and were recovered in the supernatant. Even with several optimization studies onto this capture method, it was not possible to identify a schema that could fully separate the GG microspheres from the target protein, without substantially compromising target protein yields and batch reproducibility.”
Comment 9- Line 408: “origin” should be “originate”
Authors reply: This comment was properly addressed in the manuscript. Please see page 12, line 413.
Reviewer 2 Report
Interesting research, however, some confirmation needed to improve the quality of the manuscript:
1. GG microspheres size (in micron) could be vary depend on the cation crosslinker, is it true? how it could be happen? is the stirring speed or oil type instead also affect the microsphere size?
2. Figure 1, I think we can not conclude the surface of the microsphere was smooth or rugosity, if you see the fig 1C, not all of sphere was smooth sphere.
3. From the table 1 and fig 2, what can be concluded for the bonding between GG and calcium ions? which part of hydroxyl grup of GG that could be oxidized / reduced to the bonding?
4. Reported calcium and nickel GG crosslingking together make some confusing, since some part of nickel has been reported elsewhere whereas the the application has been studied and compared in the manuscript. Do you have any important aspect to report why the two type of GG need to be compared in this manuscript?
5. The manuscript lack of conclusion
Author Response
Reviewer 2- Overall assessment of the Revised manuscript.
Interesting research, however, some confirmation needed to improve the quality of the manuscript:
Comment 1- GG microspheres size (in micron) could be vary depend on the cation crosslinker, is it true? how it could be happen? is the stirring speed or oil type instead also affect the microsphere size?
Authors reply: The authors acknowledge the Reviewer’s comment and the opportunity to clarify this issue. Indeed, it appears that different cations lead to different GG microsphere sizes. Our research group, using the same exact microsphere production method, has reported the following GG microsphere sizes for different cations (in micron) and their respective %ion integration:
Calcium (this work): 330.37 / 1.59%
Nickel (manuscript reference [35]): 239.06 / 3.08%
Magnesium (manuscript reference [35]): 300.27 / 1.04 %
Copper (manuscript reference [36]): 365.34 / 9.33%
It has been suggested that the sizes of cations can lead to different swelling properties, and ultimately, different average microsphere size [1]. One can try to derive patterns from increasing atomic or ionic radius leading to higher or lower %ion uptake in polymeric microspheres. For instance, it could be argued that increasing values of atomic radius leads to higher size GG microspheres, and that the lower sized (in atomic radius) cations would exhibit a higher %ion uptake as more of these can occupy the free space in the GG polymer network during the gelation process. Yet, when looking at the presented values, this is inconsistent with copper having the higher %uptake but also the higher average GG microsphere size, or why calcium has higher size and also higher %uptake than magnesium microspheres. Similar conclusions can be derived from trying to find the same patterns with the ionic radius. Therefore, it is not yet clear what properties of these cations lead to such distinct values. However, it appears that transition metals have higher %ion uptake levels than the corresponding alkali earth metals.
In regards to the second question, the Reviewer is correct. Indeed, stirring speed is one of the most critical production aspects that will affect final microsphere size. In the development of the emulsification procedure utilized in the present manuscript, optimization by Design of Experiments (DoE) revealed that increasing the stirring speed leads to decrease in microsphere size [2]. However, in the aforementioned GG microspheres crosslinked with the four different ions, this value was fixed at 750 rpm and would have no bearing in the final size.
[1] Bajpai, S.K.; Sharma, S. Investigation of Swelling/Degradation Behaviour of Alginate Beads Crosslinked with Ca2+ and Ba2+ Ions. React. Funct. Polym. 2004, 59, 129–140, doi:10.1016/j.reactfunctpolym.2004.01.002.
[2] Coelho, J.; Eusébio, D.; Gomes, D.; Frias, F.; Passarinha, L.A.; Sousa, Â. Biosynthesis and Isolation of Gellan Polysaccharide to Formulate Microspheres for Protein Capture. Carbohydr. Polym. 2019, 220, 236–246, doi:10.1016/j.carbpol.2019.05.011.
Comment 2- Figure 1, I think we can not conclude the surface of the microsphere was smooth or rugosity, if you see the fig 1C, not all of sphere was smooth sphere.
Authors reply: We thank the Reviewer for the pertinent comment and the opportunity to clarify. Indeed, the reviewer is correct as some microspheres appear to not be smooth in fig 1C. However, when placing the microspheres in the imaging equipment they are layered and need to be manually displaced to obtain proper snapshots. Therefore, we only considered full and intact microspheres when analyzing surface properties, as there might otherwise be the possibility that the cavities or irregularities seen in other microspheres were introduced by our manual interference and would not be representative of the real morphology. For this reason, we included approximated surface (Fig 1D) to provide a closer look at the surface of calcium-crosslinked GG microspheres. If the Reviewer compares these microspheres with the magnesium-, nickel- and copper-crosslinked microspheres provided in manuscript references [35] and [36] respectively, these snapshots are indeed smooth, by comparison. The manuscript was slightly altered to better reflect this comparison. Please see lines 136-140 in page 4.
Comment 3- From the table 1 and fig 2, what can be concluded for the bonding between GG and calcium ions? Which part of hydroxyl grup of GG that could be oxidized / reduced to the bonding?
Authors reply: The authors acknowledge the Reviewer’s questions. We can conclude that calcium is integrated in the final GG microsphere formulations at an appreciable level of 1.59%. The major calcium binding spot is coordinated by the carboxyl moieties in the GG surface. Indeed, the double helix aggregation effect that occurs in the GG gelation process, induced by divalent cations, appears to revolve around the screening of electrostatic repulsion between ionized carboxylate groups in GGs backbone as well as the formation of chemical bonds between salt ions and carboxylate groups in the glucuronate moiety [1]. Therefore the carboxyl groups at 1605 and 1743 cm-1 will be mostly responsible for establishing bonds with calcium ions. The authors have also replaced the word “bind” with “interact” when referring to hydroxyl groups as to not mislead the reader. Please see page 5, lines 166-167.
[1] Bacelar, A.H.; Silva-Correia, J.; Oliveira, J.M.; Reis, R.L. Recent Progress in Gellan Gum Hydrogels Provided by Functionalization Strategies. J Mater Chem B 2016, 4, 6164–6174, doi:10.1039/c6tb01488g.
Comment 4- Reported calcium and nickel GG crosslingking together make some confusing, since some part of nickel has been reported elsewhere whereas the the application has been studied and compared in the manuscript. Do you have any important aspect to report why the two type of GG need to be compared in this manuscript?
Authors reply: The authors acknowledge the Reviewer’s comment and the possibility to clarify and improve the manuscript. Indeed, the characterization of nickel-crosslinked GG microspheres had already been reported by our research group (refer to manuscript reference [35]) and we felt it would be redundant to “republish” near identical data. However, it was relevant to compare the differing properties of both GG microspheres, as one would consider the ~100 µm difference between microsphere sizes (nickel = 239.06 µm; calcium = 330.37 µm), and therefore the supposed larger specific surface area in the nickel-crosslinked microspheres would benefit them from an adsorption standpoint, however this was proven not to be the case. Indeed, the ionic batch method as it was described in section 2.2 was applied to both microspheres. Although the data was not shown, as other figures took precedence, we refer to this comparison at the end of section 2.2 and then again in the discussion section, respectively (lines 244-246 and lines 357-366):
“The results from the ionic exchange strategy were very similar for nickel-crosslinked microspheres, in regards to elution profiles and protein content in each batch step (data not shown).”
“Both calcium- and nickel-crosslinked GG microspheres were produced through a previously optimized W/O emulsion method, resulting in average diameters of 330.37 ± 11.38 µm and 239.06 ± 5.43 µm, respectively. These values are lower than those reported for GG microspheres produced by ionotropic gelation [45,62]. In fact, these values are approximately two-fold lower than those reported by Narkar and coworkers, using similar methodology and processing parameters [62]. Indeed, as the size of microspheres decreases it is expected an improvement in specific surface area, and consequently an improvement in adsorption capacity [63]. However, no evident change was observed between calcium and nickel microspheres after the ionic optimization batch (data not shown).”
Please note, that these comparisons were purely qualitative by SDS-PAGE/Western-Blot, as the described limitations in separating GG microspheres from “purified” samples, made it challenging to obtain feasible quantification results.
Comment 5. The manuscript lack of conclusion.
Authors reply: We are grateful for the Reviewer’s comment. Indeed, the final paragraph of the discussion was the concluding remarks and future perspectives. These have now been moved to a standalone “Conclusions” section and further future perspectives were added to provide a more complete outlook Please see page 13, lines 452-465.
Reviewer 3 Report
- I could not see some innovative information in this piece of research.
- Authors have no regard for punctuation. Many improper uses of punctuation have been found.
- Some paragraphs need to be broken for easy understanding.
- Abstract lacks data values based on the study's findings relating to different parameters.
- The method should be summarize and make it more clear.
- Mostly methodology is not cited. The author is directed to cite methods with recent studies.
- The centrifugation must be characterized by “x g” instead of “rpm”.. e.g. lines 460, 466 ..etc.
- Why is there no statistical analysis in the manuscript? Mention the method of statistical analysis in the material and methods section.
- Results of study have not been compared with the findings of earlier scientists.
- The quality of the figures must be improved. e.g. Figure 5 and 6
- Add conclusion and future perspective in one to two lines.
- Replace the references of older years with the reference of latest years.
Author Response
Reviewer 3- Overall assessment of the Revised manuscript.
Comment 1 - I could not see some innovative information in this piece of research.
Authors reply: Thanks in advance for the reviewer's comments.
Additionally, we would like to mention that the study described here applies for the first time the capture of a membrane protein, specifically human STEAP1 with high expression levels in several prostate cancers, from a simple batch method based on gellan microspheres. Coupling the batch clarified sample with a Co-immunoprecipitation polishing step yields a sample of monomeric STEAP1 with a high degree of purity. Altogether, the established reproducible strategy for the isolation of STEAP1 paves the way to gather additional insights on structural, thermal, and environmental stability characterization significantly contributing for the elucidation of the functional role and oncogenic behavior of the STEAP1 in prostate cancer microenvironment.
Comment 2 - Authors have no regard for punctuation. Many improper uses of punctuation have been found.
Authors reply: All the document was revised by a native english speaker to ensure correct spelling with the required grammar scores. Moreover, the MS word editor feature was utilized to ensure no grammatical errors were found in the main body of text.
Comment 3 - Some paragraphs need to be broken for easy understanding.
Authors reply: The authors appreciate the Reviewer’s input and the opportunity to improve the manuscript. Indeed, the Reviewer is correct. We have broken the discussion section, which corresponded to the largest wall of text, into several smaller paragraphs. These changes should help improve readability. Please see pages 11, 12 and 13.
Comment 4 - Abstract lacks data values based on the study's findings relating to different parameters.
Authors reply: The authors acknowledge the Reviewer’s comment. Due to the limitations in quantifying the output of the batch method (please refer to the reply to Reviewer’s comment 9 for further justification), it was challenging to produce reliable values, both in the main body of the manuscript and the abstract for this parameter. However, we have introduced data values related to microsphere properties, STEAP1 solubilization and optimizations to the batch method. Please see lines 21-24 in the abstract.
Comment 5 - The method should be summarize and make it more clear.
Authors reply: The authors acknowledge the Reviewer’s comment and the opportunity to clarify the manuscript. According to other Reviewer’s requests, we have made improvements in the sections relating to GG microsphere production (section 5.2, line 491), STEAP1 biosynthesis and recovery (section 5.4, lines 543-547) and to the batch method (section 5.5, lines 560-563). Please see pages 13, 14 and 15.
Comment 6 - Mostly methodology is not cited. The author is directed to cite methods with recent studies.
Authors reply: The authors acknowledge the Reviewer’s comment and the opportunity to clarify the manuscript. The gellan gum microspheres production, the mini-bioreactor STEAP1 production and recovery, the batch method, and the immunoreactive assays had been referenced in the original manuscript with the following references [73][61][35] and [74], respectively. The Co-immunoprecipitation protocol was followed according to manufacturer’s instructions, as described in lines 575-576. However, the Reviewer is correct, as we have missed the GG microsphere characterization portion. This section has now been referenced and can be seen in lines 494-495 in page 14.
Comment 7 - The centrifugation must be characterized by “x g” instead of “rpm”.. e.g. lines 460, 466 ..etc.
Authors reply: The lines identified by the Reviewer correspond to steps with agitation plates in the microsphere production steps and not centrifugations. Please see subsections 4, 5 and 6 of the materials and methods section, where all centrifugation steps were denoted in “g”.
Comment 8 - Why is there no statistical analysis in the manuscript? Mention the method of statistical analysis in the material and methods section.
Authors reply: The authors acknowledge the Reviewer’s comment. Unfortunately, no data in the manuscript requires further statistical analysis. All calculations described in the manuscript were related to means, standard deviations or simple calibration curves.
Comment 9 - Results of study have not been compared with the findings of earlier scientists.
Authors reply: The authors acknowledge the Reviewer’s pertinent question. Unfortunately, and to the best of our knowledge, it has not yet been reported a direct, STEAP1-specific quantification method. The main hinderance, especially for recombinant STEAP1, is the lack of individual enzymatic activity, which is derived from the lack of the NADPH binding domain on its structure, as opposed to other STEAP family members. In turn, this characteristic imparts limitations on the development of fast and highly sensitive analytical methods. Our research group has recently circumvented this issue, by reporting estimated values by BCA in a chromatographic setting. However, in this case this would not be possible, due to the presence of GG in the samples, which can influence absorbance values in BCA. We have stated this fact in the discussion section of the manuscript:
“However, considering the inability to completely separate GG from STEAP1, the presence of glucose moieties in the GG backbone and the fact that reducing sugars are a known interference in the BCA protein quantification assay [71], it was not possible to quantify the recovered samples. Furthermore, STEAP1 needs to form heterotrimeric ensembles with other STEAP members to attain metaloreductase activity [10], making quantification via an enzymatic iron reduction approach inaccessible, since recombinant STEAP1 was used in the batch method. Indeed, new approaches that allow the quantification of the recovered samples should be addressed in future research.”
Due to these constraints, we abstained from making quantitative comparisons to the purification results provided by other authors. However, we have introduced qualitative comparisons to other biomolecules purified/clarified through similar batch methods or comparisons to other STEAP1 purification workflows. Please see lines 422-426 in page 12 and lines 434-442 in page 13.
Comment 10 - The quality of the figures must be improved. e.g. Figure 5 and 6.
Authors reply: The authors acknowledge the reviewer’s request. The original uncropped and unedited images, as well as, the annotated images in the original resolution will be sent to the editor. The edited images will also be sent to the editor for production reasons, in the case of manuscript acceptance. The reviewer may request these original images if required.
Comment 11 - Add conclusion and future perspective in one to two lines.
Authors reply: The authors acknowledge the Reviewer’s comment and the possibility to clarify and improve the manuscript, with an issue also mentioned by Reviewer 2. Indeed, the concluding remarks and future perspectives were previously included in the last paragraph of the discussion section. They have now been moved to a standalone “Conclusions” section and further future perspectives were included to provide a more complete outlook. Please see page 13, lines 452-465.
Comment 12 - Replace the references of older years with the reference of latest years.
Authors reply: The authors acknowledge the Reviewer’s comment and the possibility to improve the manuscript. The authors have found 10 references under 2010. However, we opted to not change some of these references as they are seminal work. For instance, Hubert and coworkers discovered STEAP1 in 1999 (manuscript reference [2]), or the Laemmli method (manuscript reference [74], among others). We also believe other articles remain relevant to this day in supporting their relevant manuscript remarks (e.g. manuscript references [46] and [51]). We have however updated from Kalipatnapu, S.; Chattopadhyay; IUBMB Life 2005 to Lantez et al. Eng Life Sci 2015 and from Poetz et al. 2009 to Barroca-Ferreira et al. 2021.